# DETOX: A Redundancy-based Framework for Faster and More Robust Gradient Aggregation

**Shashank Rajput**[*]
University of Wisconsin-Madison
rajput3@wisc.edu

**Hongyi Wang**[*]
University of Wisconsin-Madison
hongyiwang@cs.wisc.edu

**Zachary Charles**
University of Wisconsin-Madison
zcharles@math.wisc.edu

**Dimitris Papailiopoulos**
University of Wisconsin-Madison
dimitris@papail.io

## Abstract

To improve the resilience of distributed training to worst-case, or Byzantine node failures, several recent approaches have replaced gradient averaging with robust aggregation methods. Such techniques can have high computational costs, often quadratic in the number of compute nodes, and only have limited robustness guarantees. Other methods have instead used redundancy to guarantee robustness, but can only tolerate limited number of Byzantine failures. In this work, we present DETOX, a Byzantine-resilient distributed training framework that combines algorithmic redundancy with robust aggregation. DETOX operates in two steps, a filtering step that uses limited redundancy to significantly reduce the effect of Byzantine nodes, and a hierarchical aggregation step that can be used in tandem with any state-of-the-art robust aggregation method. We show theoretically that this leads to a substantial increase in robustness, and has a per iteration runtime that can be nearly linear in the number of compute nodes. We provide extensive experiments over real distributed setups across a variety of large-scale machine learning tasks, showing that DETOX leads to orders of magnitude accuracy and speedup improvements over many state-of-the-art Byzantine-resilient approaches.

## 1   Introduction

To scale the training of machine learning models, gradient computations can often be distributed across multiple compute nodes. After computing these local gradients, a parameter server (PS) then averages them, and updates a global model. As the scale of data and available compute power grows, so does the probability that some compute nodes output unreliable gradients. This can be due to power outages, faulty hardware, or communication failures, or due to security issues, such as the presence of an adversary governing the output of a compute node.

Due to the difficulty in quantifying these different types of errors separately, we often model them as Byzantine failures. Such failures are assumed to be able to result in any output, adversarial or otherwise. Unfortunately, the presence of a single Byzantine compute node can result in arbitrarily bad global models when aggregating gradients via their average [1].

In distributed training, there have generally been two distinct approaches to improve Byzantine robustness. The first replaces the gradient averaging step at the PS with a robust aggregation step, such as the geometric median and variants thereof [1, 2, 3, 4, 5, 6]. The second approach instead

---

[*]Authors contributed equally to this paper and are listed alphabetically.

assigns each node redundant gradients, and uses this redundancy to eliminate the effect of Byzantine failures [7, 8, 9].

Both of the above approaches have their own limitations. For the first, robust aggregators are typically expensive to compute and scale super-linearly (in many cases quadratically [10, 4]) with the number of compute nodes. Moreover, such methods often come with limited theoretical guarantees of Byzantine robustness (*e.g.*, only establishing convergence in the limit, or only guaranteeing that the output of the aggregator has positive inner product with the true gradient [1, 10]) and often require strong assumptions, such as bounds on the dimension of the model being trained. On the other hand, redundancy or coding-theoretic based approaches offer strong or even perfect recocvery guarantees. Unfortunately, such approaches may, in the worst case, require each node to compute $\Omega(q)$ times more gradients, where $q$ is the number of Byzantine machines [7]. This overhead is prohibitive in settings with large numbers of Byzantine machines.

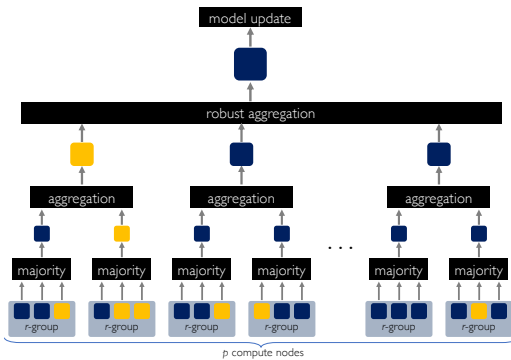

Figure 1: DETOX is a hierarchical scheme for Byzantine gradient aggregation. In its first step, the PS partitions the compute nodes in groups and assigns each node to a group with the same batch of data. After the nodes compute gradients with respect to this batch, the PS takes a majority vote of their outputs. This filters out a large fraction of the Byzantine gradients. In the second step, the PS partitions the filtered gradients in large groups, and applies a given aggregation method to each group. In the last step, the PS applies a robust aggregation method (*e.g.*, geometric median) to the previous outputs. The final output is used to perform a gradient update step.

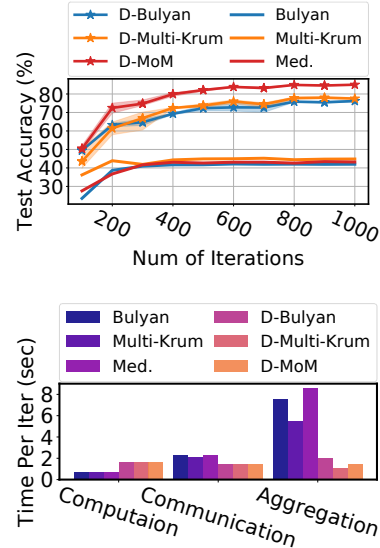

Figure 2: Top: Convergence comparisons among various vanilla robust aggregation methods and their DETOX paired versions under "a little is enough" Byzantine attack [11]. Bottom: Per iteration runtime analysis of various methods. All results are for ResNet-18 trained on CIFAR-10. The prefix "D-" stands for a robust aggregation method paired with DETOX.

**Our contributions.** In this work, we present DETOX, a Byzantine-resilient distributed training framework that first uses computational redundancy to filter out almost all Byzantine gradients, and then performs a hierarchical robust aggregation method. DETOX is scalable, flexible, and is designed to be used on top of any robust aggregation method to obtain improved robustness and efficiency. A high-level description of the hierarchical nature of DETOX is given in Fig. 1.

DETOX proceeds in three steps. First the PS partitions the compute nodes in groups of $r$ to compute the same gradients. While this step requires redundant computation at the node level, it will eventually allow for much faster computation at the PS level, as well as improved robustness. After all compute nodes send their gradients to the PS, the PS takes the majority vote of each group of gradients. We show that by setting $r$ to be logarithmic in the number of compute nodes, after the majority vote step only a constant number of Byzantine gradients are still present, even if the number of Byzantine nodes is a *constant fraction* of the total number of compute nodes. DETOX then performs hierarchical robust aggregation in two steps: First, it partitions the filtered gradients in a small number of groups, and aggregates them using simple techniques such as averaging. Second, it applies any robust aggregator (*e.g.*, geometric median [2, 6], BULYAN [10], MULTI-KRUM [4], etc.) to the averaged gradients to further minimize the effect of any remaining traces of the original Byzantine gradients.

We prove that DETOX can obtain *orders of magnitude* improved robustness guarantees compared to its competitors, and can achieve this at a nearly linear complexity in the number of compute nodes $p$, unlike methods like BULYAN [10] that require complexity that is quadratic in $p$. We extensively test our method in real distributed setups and large-scale settings, showing that by combining DETOX with previously proposed Byzantine robust methods, such as MULTI-KRUM, BULYAN, and coordinate-wise median, we increase the robustness and reduce the overall runtime of the algorithm. Moreover, we show that under strong Byzantine attacks, DETOX can lead to almost a 40% increase in accuracy over vanilla implementations of Byzantine-robust aggregation. A brief performance comparison with some of the current state-of-the-art aggregators in shown in Fig. 2.

**Related work.** The topic of Byzantine fault tolerance has been extensively studied since the early 80s by Lamport et al. [12], and deals with worst-case, and/or adversarial failures, *e.g.*, system crashes, power outages, software bugs, and adversarial agents that exploit security flaws. In the context of distributed optimization, these failures are manifested through a subset of compute nodes returning to the master flawed or adversarial updates. It is now well understood that first-order methods, such as gradient descent or mini-batch SGD, are not robust to Byzantine errors; even a single erroneous update can introduce arbitrary errors to the optimization variables.

Byzantine-tolerant ML has been extensively studied in recent years [13, 14, 15, 16, 17, 2], establishing that while average-based gradient methods are susceptible to adversarial nodes, median-based update methods can in some cases achieve better convergence, while being robust to some attacks. Although theoretical guarantees are provided in many works, the proposed algorithms in many cases only ensure a weak form of resilience against Byzantine failures, and often fail against strong Byzantine attacks [10]. A stronger form of Byzantine resilience is desirable for most of distributed machine learning applications. To the best of our knowledge, DRACO [7] and BULYAN [10] are the only proposed methods that guarantee strong Byzantine resilience. However, as mentioned above, DRACO requires heavy redundant computation from the compute nodes, while BULYAN requires heavy computation overhead on the PS end.

We note that [18] presents an alternative approach that does not fit easily under either category, but requires convexity of the underlying loss function. Finally, [19] examines the robustness of SIGNSGD with a majority vote aggregation, but study a restricted Byzantine failure setup that only allows for a blind multiplicative adversary.

## 2 Problem Setup

Our goal is to solve solve the following empirical risk minimization problem: $\min_w F(w) := \frac{1}{n} \sum_{i=1}^n f_i(w)$ where $w \in \mathbb{R}^d$ denotes the parameters of a model, and $f_i$ is the loss function on the $i$-th training sample. To approximately solve this problem, we often use mini-batch SGD. First, we initialize at some $w_0$. At iteration $t$, we sample $S_t$ uniformly at random from $\{1, \ldots, n\}$, and then update via

$$w_{t+1} = w_t - \frac{\eta_t}{|S_t|} \sum_{i \in S_t} \nabla f_i(w_t), \tag{1}$$

where $S_t$ is a randomly selected subset of the $n$ data points. To perform mini-batch SGD in a distributed manner, the global model $w_t$ is stored at the PS and updated according to (1), *i.e.*, by using the mean of gradients that are evaluated at the compute nodes.

Let $p$ denote the total number of compute nodes. At each iteration $t$, during distributed mini-batch SGD, the PS broadcasts $w_t$ to each compute node. Each compute node is assigned $S_{i,t} \subseteq S_t$, and then evaluates the mean of gradients $g_i = \frac{1}{|S_{i,t}|} \sum_{j \in S_{i,t}} \nabla f_j(w_t)$. The PS then updates the global model via $w_{t+1} = w_t - \frac{\eta_t}{p} \sum_{i=1}^p g_i$. We note that in our setup we assume that the PS is the owner of the data, and has access to the entire data set of size $n$.

**Distributed training with Byzantine nodes** We assume that a fixed subset $Q$ of size $q$ of the $p$ compute nodes are Byzantine. Let $\hat{g}_i$ be the output of node $i$. If $i$ is not Byzantine ($i \notin Q$), we say it is "honest", in which case its output $\hat{g}_i = g_i$ where $g_i$ is the true mean of gradients assigned to node $i$. If $i$ is Byzantine ($i \in Q$), its output $\hat{g}_i$ can be any $d$-dimensional vector. The PS receives $\{\hat{g}_i\}_{i=1}^p$, and can then process these vectors to produce some approximation to the true gradient update in (1).

We make no assumptions on the Byzantine outputs. In particular, we allow adversaries with full information about $F$ and $w_t$, and that the Byzantine compute nodes can collude. Let $\epsilon = q/p$ be the fraction of Byzantine nodes. We will assume $\epsilon < 1/2$ throughout.

# 3 DETOX: A Redundancy Framework to Filter most Byzantine Gradients

We now describe DETOX, a framework for Byzantine-resilient mini-batch SGD with $p$ nodes, $q$ of which are Byzantine. Let $b \geq p$ be the desired batch-size, and let $r$ be an odd integer. We refer to $r$ as the *redundancy ratio*. For simplicity, we will assume $r$ divides $p$ and that $p$ divides $b$. DETOX can be directly extended to the setting where this does not hold.

DETOX first computes a random partition of $[p]$ in $p/r$ node groups $A_1, \ldots, A_{p/r}$ each of size $r$. This will be fixed throughout. We then initialize at some $w_0$. For $t \geq 0$, we wish to compute some approximation to the gradient update in (1). To do so, we need a Byzantine-robust estimate of the true gradient. Fix $t$, and let us suppress the notation $t$ when possible. As in mini-batch SGD, let $S$ be a subset of $[n]$ of size $b$, with each element sampled uniformly at random from $[n]$. We then partition of $S$ in groups $S_1, \ldots, S_{p/r}$ of size $br/p$. For each $i \in A_j$, the PS assigns node $i$ the task of computing

$$g_j := \frac{1}{|S_j|} \sum_{k \in S_j} \nabla f_k(w) = \frac{p}{rb} \sum_{k \in S_j} \nabla f_k(w). \tag{2}$$

If $i$ is an honest node, then its output is $\hat{g}_i = g_j$, while if $i$ is Byzantine, it outputs some $d$-dimensional $\hat{g}_i$, which is then sent to the PS. The PS then computes $z_j := \text{maj}(\{\hat{g}_i | i \in A_j\})$, where $\text{maj}$ denotes the majority vote. If there is no majority, we set $z_j = 0$. We will refer to $z_j$ as the "vote" of group $j$.

Since some of these votes are still Byzantine, we must do some robust aggregation of the vote. We employ a hierarchical robust aggregation process HIER-AGGR, which uses two user-specified aggregation methods $\mathcal{A}_0$ and $\mathcal{A}_1$. First, the votes are partitioned in to $k$ groups. Let $\hat{z}_1, \ldots, \hat{z}_k$ denote the output of $\mathcal{A}_0$ on each group. The PS then computes $\hat{G} = \mathcal{A}_1(\hat{z}_1, \ldots, \hat{z}_k)$ and updates the model via $w = w - \eta \hat{G}$. This hierarchical aggregation resembles a median of means approach on the votes [20], and has the benefit of improved robustness and efficiency. We discuss this in further detail in Section 4. A description of DETOX is given in Algorithm 1.

---

**Algorithm 1** DETOX: Algorithm to be performed at the parameter server

---

**input** Batch size $b$, redundancy ratio $r$, compute nodes $1, \ldots, p$, step sizes $\{\eta_t\}_{t \geq 0}$.
 1: Randomly partition $[p]$ in "node groups" $\{A_j | 1 \leq j \leq p/r\}$ of size $r$.
 2: **for** $t = 0$ **to** $T$ **do**
 3:     Draw $S_t$ of size $b$ randomly from $[n]$.
 4:     Partition $S_t$ in to groups $\{S_{t,j} | 1 \leq j \leq p/r\}$ of size $rb/p$.
 5:     For each $j \in [p/r], i \in A_j$, push $w_t$ and $S_{t,j}$ to compute node $i$.
 6:     Receive the (potentially Byzantine) $p$ gradients $\hat{g}_{t,i}$ from each node.
 7:     Let $z_{t,j} := \text{maj}(\{\hat{g}_{t,i} | i \in A_j\})$, and 0 if no majority exists.     %Filtering step
 8:     Set $\hat{G}_t = \text{HIER-AGGR}(\{z_{t,1}, \ldots, z_{t,p/r}\})$.     %Hierarchical aggregation
 9:     Set $w_{t+1} = w_t - \eta \hat{G}_t$.     %Gradient update
10: **end for**

---

**Algorithm 2** HIER-AGGR: Hierarchical aggregation

---

**input** Aggregators $\mathcal{A}_0, \mathcal{A}_1$, votes $\{z_1, \ldots, z_{p/r}\}$, vote group size $k$.
 1: Let $\hat{p} := p/r$.
 2: Randomly partition $\{z_1, \ldots, z_{\hat{p}}\}$ in to $k$ "vote groups" $\{Z_j | 1 \leq j \leq k\}$ of size $\hat{p}/k$.
 3: For each vote group $Z_j$, calculate $\hat{z}_j = \mathcal{A}_0(Z_j)$.
 4: Return $\mathcal{A}_1(\{\hat{z}_1, \ldots, \hat{z}_k\})$.

---

## 3.1 Filtering out Almost Every Byzantine Node

We now show that DETOX filters out the vast majority of Byzantine gradients. Fix the iteration $t$. Recall that all honest nodes in a node group $A_j$ send $\hat{g}_j = g_j$ as in (2) to the PS. If $A_j$ has more

honest nodes than Byzantine nodes then $z_j = g_j$ and we say $z_j$ is honest. If not, then $z_j$ may not equal $g_j$ in which case $z_j$ is a Byzantine vote. Let $X_j$ be the indicator variable for whether block $A_j$ has more Byzantine nodes than honest nodes, and let $\hat{q} = \sum_j X_j$. This is the number of Byzantine votes. By filtering, DETOX goes from a Byzantine compute node ratio of $\epsilon = q/p$ to a Byzantine vote ratio of $\hat{\epsilon} = \hat{q}/\hat{p}$ where $\hat{p} = p/r$.

We first show that $\mathbb{E}[\hat{q}]$ decreases *exponentially* with $r$, while $\hat{p}$ only decreases linearly with $r$. That is, by incurring a constant factor loss in compute resources, we gain an exponential improvement in the reduction of Byzantine nodes. Thus, even small $r$ can drastically reduce the Byzantine ratio of votes. This observation will allow us to instead use robust aggregation methods on the $z_j$, *i.e.*, the votes, greatly improving our Byzantine robustness. We have the following theorem about $\mathbb{E}[\hat{q}]$. All proofs can be found in the appendix. Note that throughout, we did not focus on optimizing constants.

**Theorem 1.** *There is a universal constant $c$ such that if the fraction of Byzantine nodes is $\epsilon < c$, then the effective number of Byzantine votes after filtering satisfies $\mathbb{E}[\hat{q}] = \mathcal{O}\left(\epsilon^{(r-1)/2} q/r\right)$.*

We now wish to use this to derive high probability bounds on $\hat{q}$. While the variables $X_i$ are not independent, they are negatively correlated. By using a version of Hoeffding's inequality for weakly dependent variables, we can show that if the redundancy is logarithmic, *i.e.*, $r \approx \log(q)$, then with high probability the number of effective Byzantine votes drops to a constant, *i.e.*, $\hat{q} = \mathcal{O}(1)$.

**Corollary 2.** *There is a constant $c$ such that if and $\epsilon \leq c$ and $r \geq 3 + 2\log_2(q)$ then for any $\delta \in (0, \frac{1}{2})$, with probability at least $1 - \delta$, we have that $\hat{q} \leq 1 + 2\log(1/\delta)$.*

In the next section, we exploit this dramatic reduction of Byzantine votes to derive strong robustness guarantees for DETOX.

## 4 DETOX Improves the Speed and Robustness of Robust Estimators

Using the results of the previous section, if we set the redundancy ratio to $r \approx \log(q)$, the filtering stage of DETOX reduces the number of Byzantine votes $\hat{q}$ to roughly a constant. While we could apply some robust aggregator $\mathcal{A}$ directly to the output votes of the filtering stage, such methods often scale poorly with the number of votes $\hat{p}$. By instead applying HIER-AGGR, we greatly improve efficiency and robustness. Recall that in HIER-AGGR, we partition the votes into $k$ "vote groups", apply some $\mathcal{A}_0$ to each group, and apply some $\mathcal{A}_1$ to the $k$ outputs of $\mathcal{A}_0$. We analyze the case where $k$ is roughly constant, $\mathcal{A}_0$ computes the mean of its inputs, and $\mathcal{A}_1$ is a robust aggregator. In this case, HIER-AGGR is analogous to the Median of Means (MoM) method from robust statistics [20].

**Improved speed.** Suppose that without redundancy, the time required for the compute nodes to finish is $T$. Applying KRUM [1], MULTI-KRUM [4], and BULYAN [10] to their $p$ outputs requires $\mathcal{O}(p^2 d)$ operations, so their overall runtime is $\mathcal{O}(T + p^2 d)$. In DETOX, the compute nodes require $r$ times more computation to evaluate redundant gradients. If $r \approx \log(q)$, this can be done in $\mathcal{O}(\ln(q)T)$. With HIER-AGGR as above, DETOX performs three major operations: (1) majority voting, (2) mean computation of the $k$ vote groups and (3) robust aggregation of the these $k$ means using $\mathcal{A}_1$. (1) and (2) require $\mathcal{O}(pd)$ time. For practical $\mathcal{A}_1$ aggregators, including MULTI-KRUM and BULYAN, (3) requires $\mathcal{O}(k^2 d)$ time. Since $k \ll p$, DETOX has runtime $\mathcal{O}(\ln(q)T + pd)$. If $T = \mathcal{O}(d)$ (which generally holds for gradient computations), KRUM, MULTI-KRUM, and BULYAN require $\mathcal{O}(p^2 d)$ time, but DETOX only requires $\mathcal{O}(pd)$ time. Thus, DETOX can lead to significant speedups, especially when the number of workers is large.

**Improved robustness.** To analyze robustness, we first need some distributional assumptions. At a given iteration, let $G$ denote the full gradient of $F(w)$. Throughout this section, we assume that the gradient of each sample is drawn from a distribution $\mathcal{D}$ on $\mathbb{R}^d$ with mean $G$ and covariance $\Sigma$. Let $\sigma^2 = \text{Tr}(\Sigma)$, we'll refer to this as variance. In DETOX, the "honest" votes $z_i$ will also have mean $G$, but their variance will be $\sigma^2 p/rb$. This is because each honest compute node gets $rb/p$ samples, so its variance is reduced by $rb/p$. Note that this variance reduction is integral in proving that we achieve optimal rates (see Theorem 3 and the discussion after it). To see this intuitively, consider a scenario without Byzantine machines, then the variance of empirical mean is $\sigma^2/b$. A simple calculation shows that variance of the mean of each "vote group" is $\frac{\sigma^2 p/rb}{\hat{p}/k} = k\sigma^2/b$ where $k$ is the number of vote groups. Thus, if $k$ is small, we are still able to optimally reduce the variance.

Suppose $\hat{G}$ is some approximation to the true gradient $G$. We say that $\hat{G}$ is a $\Delta$-inexact gradient oracle for $G$ if $\|\hat{G} - G\| \leq \Delta$. [5] shows that access to a $\Delta$-inexact gradient oracle is sufficient to upper bound the error of a model $\hat{w}$ produced by performing gradient updates with $\hat{G}$. Thus, to bound the robustness of an aggegator, it suffices to bound $\Delta$. Under the distributional assumptions above, we will derive bounds on $\Delta$ for the hierarchical aggregator $\mathcal{A}$ with different base aggregators $\mathcal{A}_1$.

We will analyze DETOX when $\mathcal{A}_0$ computes the mean of the vote groups, and $\mathcal{A}_1$ is geometric median, coordinate-wise median, or $\alpha$-trimmed mean [6]. We will denote the approximation $\hat{G}$ to $G$ computed by DETOX in these three instances by $\hat{G}_1, \hat{G}_2$ and $\hat{G}_3$, respectively. Using the proof techniques similar to [20], we get the following.

**Theorem 3.** *Assume $r \geq 3 + 2\log_2(q)$ and $\epsilon \leq c$ where $c$ is the constant from Corollary 2. There are constants $c_1, c_2, c_3$ such that for all $\delta \in (0, 1/2)$, with probability at least $1 - 2\delta$:*

1. *If $k = 128\ln(1/\delta)$, then $\hat{G}_1$ is a $c_1\sigma\sqrt{\ln(1/\delta)/b}$-inexact gradient oracle.*

2. *If $k = 128\ln(d/\delta)$, then $\hat{G}_2$ is a $c_2\sigma\sqrt{\ln(d/\delta)/b}$-inexact gradient oracle.*

3. *If $k = 128\ln(d/\delta)$ and $\alpha = \frac{1}{4}$, then $\hat{G}_3$ is a $c_3\sigma\sqrt{\ln(d/\delta)/b}$-inexact gradient oracle.*

The above theorem has three important implications. First, we can derive robustness guarantees for DETOX that are virtually independent of the Byzantine ratio $\epsilon$. Second, even when there are no Byzantine machines, it is known that no aggregator can achieve $\Delta = o(\sigma/\sqrt{b})$ [21], and because we achieve $\Delta = \tilde{O}(\sigma/\sqrt{b})$, we cannot expect to get an order of better robustness by any other aggregator. Third, other than a logarithmic dependence on $q$, there is no dependence on the number of nodes $p$. Even as $p$ and $q$ increase, we still maintain roughly the same robustness guarantees.

By comparison, the robustness guarantees of KRUM and Geometric Median applied directly to the compute nodes worsens as as $p$ increases [17, 3]. Similarly, [6] show if we apply coordinate-wise median to $p$ nodes, each of which are assigned $b/p$ samples, we get a $\Delta$-inexact gradient oracle where $\Delta = \mathcal{O}(\sigma\sqrt{\epsilon p/b} + \sigma\sqrt{d/b})$. If $\epsilon$ is constant and $p$ is comparable to $b$, then this is roughly $\sigma$, whereas DETOX can produce a $\Delta$-inexact gradient oracle for $\Delta = \tilde{\mathcal{O}}(\sigma/\sqrt{b})$. Thus, the robustness of DETOX can scale much better with the number of nodes than naive robust aggregation of gradients.

## 5 Experiments

In this section we present an experimental study on pairing DETOX with a set of previously proposed robust aggregation methods, including MULTI-KRUM [17], BULYAN [10], coordinate-wise median [5]. We also incorporate DETOX with a recently proposed Byzantine resilient distributed training method *i.e.* SIGNSGD with majority vote [19]. We conduct extensive experiments on the scalability and robustness of these Byzantine-resilient methods, and the improvements gained when pairing them with DETOX. All our experiments are deployed on real distributed clusters under various Byzantine attack models. Our implementation is publicly available for reproducibility [2].

### 5.1 Experimental Setup

The main findings are as follows: 1) Applying DETOX leads to significant speedups, *e.g.*, up to an order of magnitude end-to-end training speedup is observed; 2) in defending against state-of-the-art Byzantine attacks, DETOX leads to significant Byzantine-resilience improvement, *e.g.*, applying BULYAN on top of DETOX improves the test-set prediction accuracy from 11% to 60% when training VGG13-BN on CIFAR-100 under the "a little is enough" (ALIE) [11] Byzantine attack. Moreover, incorporating SIGNSGD with DETOX improves the test set prediction accuracy from $34.92\%$ to $78.75\%$ when defending against a *constant Byzantine attack* for ResNet-18 trained on CIFAR-10.

We implemented vanilla versions of the aforementioned Byzantine resilient methods, as well as versions of these methods pairing with DETOX, in PyTorch [22] with MPI4py [23]. Our experiments are deployed on a cluster of 46 m5.2xlarge instances on Amazon EC2, where 1 node serves as the PS and the remaining $p = 45$ nodes are compute nodes. In all the following experiments, we set the number of Byzantine nodes to be $q = 5$. We also study the performance of all considered methods with smaller number (and without) Byzantine nodes, the result can be found in the Appendix B.6.

## 5.2 Implementation of DETOX

In DETOX, the 45 compute nodes are randomly partitioned into node groups of size $r = 3$, which gives $p/r = 15$ node groups. Batch size $b$ is set to $1,440$. In each iteration of the vanilla Byzantine resilient methods, each compute node evaluates $b/p = 32$ gradients sampled from its partition of data while in DETOX each node evaluates $r\times$ more gradients *i.e.* $rb/p = 96$, which makes DETOX $r\times$ more computationally expensive than the vanilla Byzantine resilient methods. Compute nodes in the same node group evaluate the same gradients to create algorithmic redundancy for the majority voting stage in DETOX. The mean of these locally computed gradients is sent back to the PS. Note that although DETOX requires each compute node evaluate $r\times$ more gradients, the communication cost of DETOX is the same as the vanilla Byzantine resilient methods since only the gradient means are communicated instead of individual gradients. After receiving all gradient means from the compute nodes, the PS uses either vanilla Byzantine-resilient methods or their DETOX paired variants.

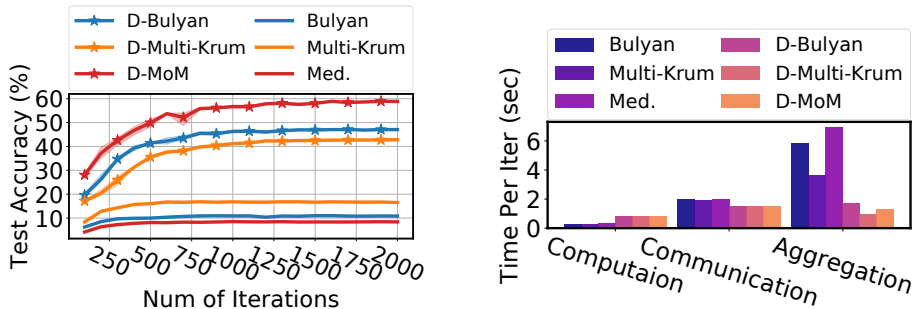

Figure 3: Results of VGG13-BN on CIFAR-100. Left: Convergence performance of various robust aggregation methods against ALIE attack. Right: Per iteration runtime analysis of various robust aggregation methods.

We emphasize that DETOX is not simply a new robust aggregation technique. It is instead a general Byzantine-resilient distributed training framework, and any robust aggregation method can be immediately implemented on top of it to increase its Byzantine-resilience and scalability. Note that after the majority voting stage on the PS one has a wide range of choices for $\mathcal{A}_0$ and $\mathcal{A}_1$. In our implementations, we had the following setups: 1) $\mathcal{A}_0 = \text{Mean}$, $\mathcal{A}_1 = \text{Coordinate-size Median}$, 2) $\mathcal{A}_0 = \text{MULTI-KRUM}$, $\mathcal{A}_1 = \text{Mean}$, 3) $\mathcal{A}_0 = \text{BULYAN}$, $\mathcal{A}_1 = \text{Mean}$, and 4) $\mathcal{A}_0 = \text{coordinate-wise}$ majority vote, $\mathcal{A}_1 = \text{coordinate-wise majority vote}$ (designed specifically for pairing DETOX with SIGNSGD). We tried $\mathcal{A}_0 = \text{Mean}$ and $\mathcal{A}_1 = \text{MULTI-KRUM/BULYAN}$ but we found that setups 2) and 3) had better resilience than these choices. More details on the implementation and system-level optimizations that we performed can be found in the Appendix B.1.

**Byzantine attack models**   We consider two Byzantine attack models for pairing MULTI-KRUM, BULYAN, and coordinate-wise median with DETOX. First, we consider the *"reversed gradient"* attack, where Byzantine nodes that were supposed to send $\mathbf{g} \in \mathbb{R}^d$ to the PS instead send $-c\mathbf{g}$, for some $c > 0$. Secondly, we study the recently proposed ALIE [11] attack, where the Byzantine compute nodes collude and use their locally calculated gradients to estimate the coordinate-wise mean and standard deviation of the entire set of gradients of all other compute nodes. The Byzantine nodes then use the estimated mean and variance to manipulate the gradient they send back to the PS. To be more specific, Byzantine nodes will send $\hat{\mu}_i + z \cdot \hat{\sigma}_i, \forall i \in [d]$ where $\hat{\mu}$ and $\hat{\sigma}$ are the estimated coordinate-wise mean and standard deviation each gradient dimension and $z$ is a hyper-parameter which was tuned empirically in [11]. Finally, to compare the resilience of the vanilla SIGNSGD and the one paired with DETOX, we consider the *"constant Byzantine attack"* where Byzantine compute nodes send a constant gradient matrix with dimension same as that of the true gradient but all elements set to $-1$.

**Datasets and models**   Our experiments are over ResNet-18 [24] on CIFAR-10 and VGG13-BN [25] on CIFAR-100. For each dataset, we use data augmentation (random crops, and flips) and image normalization. Also, we tune the learning rate schedules and use the constant momentum at $0.9$ in all experiments. The details of parameter tuning and dataset normalization are in the Appendix B.2.

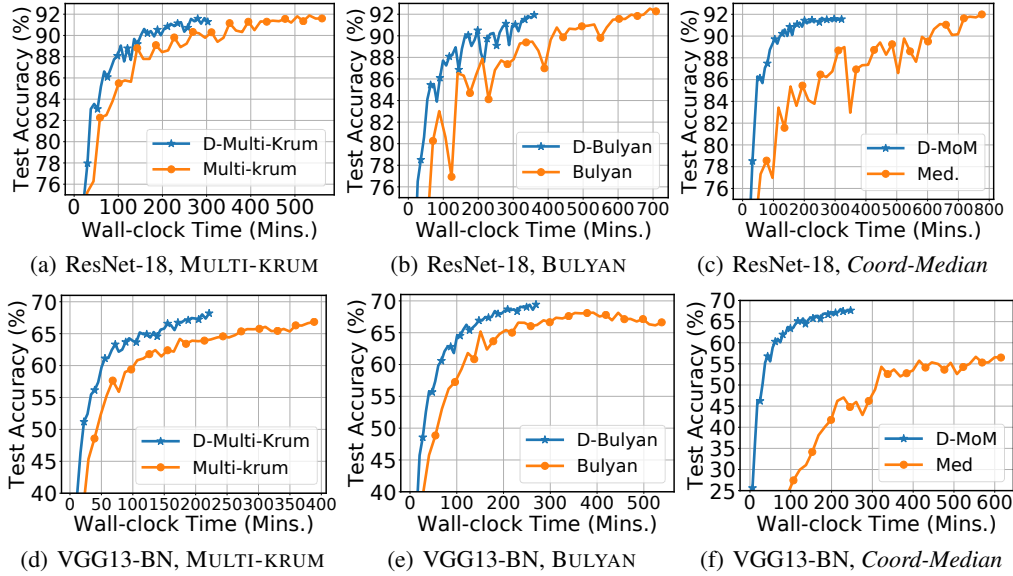

Figure 4: End-to-end comparisons between DETOX paired with different baseline methods under *reverse gradient* attack. (a)-(c): Vanilla vs. DETOX paired version of MULTI-KRUM, BULYAN, and coordinate-wise median on ResNet-18 trained on CIFAR-10. (d)-(f): Same comparisons for VGG13-BN trained on CIFAR-100.

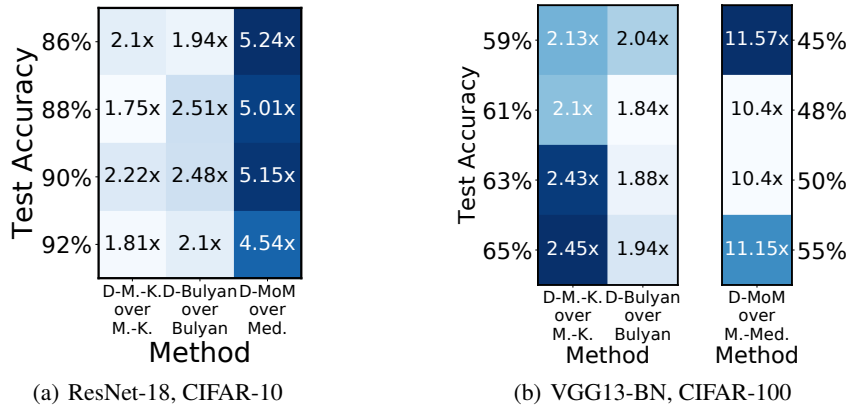

Figure 5: Speedups in converging to given accuracies for vanilla robust aggregation methods and their DETOX-paired variants under *reverse gradient* attack: (a) ResNet-18 on CIFAR-10, (b) VGG13-BN on CIFAR-100

## 5.3 Results

**Scalability** We report a per-iteration runtime of all considered robust aggregations and their DETOX paired variants on both CIFAR-10 over ResNet-18 and CIFAR-100 over VGG-13. The results on ResNet-18 and VGG13-BN are shown in Figure 2 and 3. We observe that although DETOX requires slightly more compute time per iteration, due to its algorithmic redundancy as explained in Section 5.2, it largely reduces the PS computation cost during the aggregation stage, which matches our theoretical analysis. Surprisingly, we observe that by applying DETOX, the communication costs decrease. This is because the variance of computation time among compute nodes increases with heavier computational redundancy. Therefore, after applying DETOX, compute nodes tend not to send their gradients to the PS at the same time, which mitigates a potential network bandwidth congestion. In a nutshell, applying DETOX can lead to up to $3\times$ per-iteration speedup.

**Byzantine-resilience under various attacks** We first study the Byzantine-resilience of all considered methods under the ALIE attack, which to the best of our knowledge, is the strongest Byzantine attack proposed in the literature. The results on ResNet-18 and VGG13-BN are shown in Figure 2 and 3 respectively. Applying DETOX leads to significant improvement in Byzantine-resilience

Table 1: Defense results summary for ALIE attacks [11]; the reported numbers are test set prediction accuracy.

| | D-Multi-krum | D-Bulyan | **D-Med.** | Multi-krum | Bulyan | Med. |
|---|---|---|---|---|---|---|
| ResNet-18 | 80.3% | 76.8% | **86.21%** | 45.24% | 42.56% | 43.7% |
| VGG13-BN | 42.98% | 46.82% | **59.51%** | 17.18% | 11.06% | 8.64% |

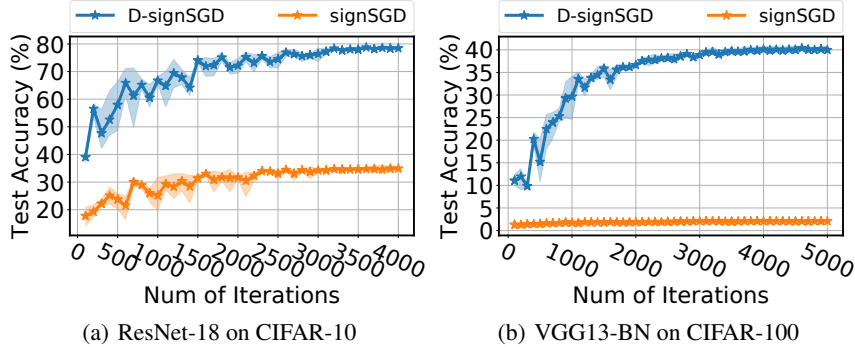

(a) ResNet-18 on CIFAR-10      (b) VGG13-BN on CIFAR-100

Figure 6: Convergence comparisons between DETOX paired with SIGNSGD and vanilla SIGNSGD under *constant Byzantine attack* on: (a) ResNet-18 trained on CIFAR-10; (b) VGG13-BN trained on CIFAR-100

compared to vanilla MULTI-KRUM, BULYAN, and coordinate-wise median on both datasets as shown in Table 1. We then consider the *reverse gradient* attack, the results are shown in Figure 4. Since *reverse gradient* is a much weaker attack, all vanilla robust aggregation methods and their DETOX paired variants defend well. Moreover, applying DETOX leads to significant end-to-end speedups. In particular, combining the coordinate-wise median with DETOX led to a $5\times$ speedup gain in the amount of time to achieve to 90% test set prediction accuracy for ResNet-18 trained on CIFAR-10. The speedup results are shown in Figure 5. For VGG13-BN trained on CIFAR-100, an order of magnitude end-to-end speedup can be observed in coordinate-wise median applied on top of DETOX.

**Comparison between DETOX and SIGNSGD** We compare DETOX paired SIGNSGD with vanilla SIGNSGD where only the sign of each gradient coordinate is sent to the PS. The PS, on receiving these gradient signs, takes coordiante-wise majority votes to get the model update. We consider a stronger *constant Byzantine attack* introduced in Section 5.2. The details of our implementation and hyper-parameters used are in Appendix B.4. The results on both the considered datasets are shown in Figure 6 where we see that DETOX paired with SIGNSGD improves the Byzantine resilience of SIGNSGD significantly. For ResNet-18 trained on CIFAR-10, DETOX improves testset prediction accuracy of vanilla SIGNSGD from $34.92\%$ to $78.75\%$; while for VGG13-BN trained on CIFAR-100, DETOX improves testset prediction accuracy (TOP-1) of vanilla SIGNSGD from $2.12\%$ to $40.37\%$.

For completeness, we compare DETOX with DRACO [7]. This is not the focus of this work, as we are primarily interested in showing that DETOX improves the robustness of traditional robust aggregators. However the comparisons with DRACO are in Appendix B.7. Another experimental study of mean estimation task over synthetic data that directly matches our theory can be found in Appendix B.5.

## 6 Conclusion

In this paper, we present DETOX, a new framework for Byzantine-resilient distributed training. Notably, any robust aggregator can be immediatley used with DETOX to increase its robustness and efficiency. We demonstrate these improvements theoretically and empirically. In the future, we would like to devise a privacy-preserving version of DETOX, as currently it requires the PS to be the owner of the data, and also to partition data among compute nodes, which hurts the data privacy. Overcoming this limitation would allow us to develop variants of DETOX for federated learning.

## Acknowledgments

This research is supported by an NSF CAREER Award #1844951, a Sony Faculty Innovation Award, an AFOSR & AFRL Center of Excellence Award FA9550-18-1-0166, and an NSF TRIPODS Award #1740707. The authors also thank Ankit Pensia for useful discussions about the Median of Means approach.

## Footnotes

[2]https://github.com/hwang595/DETOX

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
