[Supplementary Material]

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

# A  Proofs

## A.1  Proof of Theorem 1

The following is a more precise statement of the theorem.

**Theorem.** *If $r > 3$, $p \geq 2r$ and $\epsilon < 1/40$ then $\mathbb{E}[\hat{q}]$ falls as $\mathcal{O}\left(q(40\epsilon(1-\epsilon))^{(r-1)/2}/r\right)$ which is exponential in r.*

*Proof.* By direct computation,

$$
\mathbb{E}(\hat{q}) = \mathbb{E}\left(\sum_{i=1}^{p/r} X_i\right)
$$

$$
= \frac{p}{r}\mathbb{E}(X_i)
$$

$$
= \frac{p}{r}\frac{\sum_{i=0}^{(r-1)/2}\binom{q}{r-i}\binom{p-q}{i}}{\binom{p}{r}}
$$

$$
\leq \frac{p}{r}\frac{\frac{r+1}{2}\binom{q}{(r+1)/2}\binom{p-q}{(r-1)/2}}{\binom{p}{r}}
$$

$$
\leq \frac{p}{r}\frac{r+1}{2}\frac{\binom{r}{(r-1)/2}q^{(r+1)/2}(p-q)^{(r-1)/2}}{(p-r)^r}
$$

$$
= \frac{p}{r}\frac{r+1}{2}\frac{\binom{r}{(r-1)/2}q^{(r+1)/2}(p-q)^{(r-1)/2}}{p^r(1-r/p)^r}
$$

$$
\leq \frac{p}{r}\frac{r+1}{2}\frac{\binom{r}{(r-1)/2}q^{(r+1)/2}(p-q)^{(r-1)/2}}{p^r(1/2)^r}
$$

$$
= \frac{p}{r}(r+1)2^{r-1}\binom{r}{(r-1)/2}\epsilon^{(r+1)/2}(1-\epsilon)^{(r-1)/2}.
$$

Note that $\binom{r}{(r-1)/2}$ is the coefficient of $x^{(r+1)/2}(1-x)^{(r-1)/2}$ in the binomial expansion of $1 = 1^r = (x+(1-x))^r$. Therefore, setting $x = \frac{1}{2}$, we find that $\binom{r}{(r-1)/2} \leq 2^r$. Therefore,

$$
\frac{p}{r}(r+1)2^{r-1}\binom{r}{(r-1)/2}\epsilon^{(r+1)/2}(1-\epsilon)^{(r-1)/2}
$$

$$
\leq \frac{p}{r}(r+1)2^{2r-1}\epsilon^{(r+1)/2}(1-\epsilon)^{(r-1)/2}
$$

$$
= \frac{p}{r}(r+1)\epsilon\left(2^{2r-1}\epsilon^{(r-1)/2}(1-\epsilon)\right)^{(r-1)/2}
$$

$$
= \frac{2q}{r}(r+1)\left(16\epsilon(1-\epsilon)\right)^{(r-1)/2}
$$

$$
= \frac{2q}{r}\left(16(r+1)^{2/(r-1)}\epsilon(1-\epsilon)\right)^{(r-1)/2} \qquad .
$$

Note that since $r > 3$ and $r$ is odd, we have $r \geq 5$. Therefore,
$$\mathbb{E}(\hat{q}) \leq 2q(40\epsilon(1-\epsilon))^{(r-1)/2}/r.$$

$\square$

For $r = 3$, we have the following lemma.

**Lemma 4.** *If $r = 3$, then $\mathbb{E}[\hat{q}] \leq q(4\delta - 2\delta^2)/3$ when $n \geq 6$.*

*Proof.*

$$\mathbb{E}(q_e) = \mathbb{E}(\sum_{i=1}^{\frac{p}{3}} X_i) = \frac{p}{3} E(X_i) = \frac{p}{3} \frac{\binom{q}{3} + \binom{q}{2}\binom{p-q}{1}}{\binom{n}{3}}$$

$$= \frac{p}{3} \frac{q(q-1)(3p-2q-2)}{p(p-1)(p-2)} = \frac{q}{3} \frac{\left(\epsilon - \frac{1}{p}\right)\left(3 - 2\delta - \frac{2}{p}\right)}{\left(1 - \frac{1}{p}\right)\left(1 - \frac{2}{p}\right)}$$

$$\leq \frac{q}{3} \epsilon \frac{3 - 2\epsilon - \frac{2}{p}}{1 - \frac{2}{p}} \leq q\epsilon(4 - 2\epsilon)/3$$

$\square$

## A.2 Proof of Corollary 2

From Theorem 1 we see that $\mathbb{E}[\hat{q}] \leq 2q(40\epsilon(1-\epsilon))^{(r-1)/2}/r \leq 2q(40\epsilon)^{(r-1)/2}$. Now, straightforward analysis implies that if $\epsilon \leq 1/80$ and $r \geq 3 + 2\log_2 q$ then $\mathbb{E}[\hat{q}] \leq 1$. We will then use the following Lemma:

**Lemma 5.** *For all $\theta > 0$,*

$$\mathbb{P}\left[\hat{q} \geq \mathbb{E}[\hat{q}](1 + \theta)\right] \leq \left(\frac{1}{1 + \theta/2}\right)^{\mathbb{E}[\hat{q}]\theta/2}$$

Now, using Lemma 5 and assuming $\theta \geq 2$,

$$\mathbb{P}\left[\hat{q} \geq \mathbb{E}[\hat{q}](1 + \theta)\right] \leq \left(\frac{1}{1 + \theta/2}\right)^{\mathbb{E}[\hat{q}]\theta/2}$$

$$\implies \mathbb{P}\left[\hat{q} \geq 1 + \mathbb{E}[\hat{q}]\theta\right] \leq \left(\frac{1}{1 + \theta/2}\right)^{\mathbb{E}[\hat{q}]\theta/2}$$

$$\implies \mathbb{P}\left[\hat{q} \geq 1 + \mathbb{E}[\hat{q}]\theta\right] \leq 2^{-\mathbb{E}[\hat{q}]\theta/2}$$

where we used the fact that $\mathbb{E}[\hat{q}] \leq 1$ in the first implication and the assumption that $\theta \geq 2$ in the second. Setting $\delta := 2^{-\mathbb{E}[\hat{q}]\theta/2}$, we get the probability bound. Finally, setting $\delta \leq 1/2$ makes $\theta \geq 2$, which completes the proof.

## A.3 Proof of Lemma 5

We will prove the following:

$$P\left[\hat{q} \geq \mathbb{E}[\hat{q}](1 + \theta)\right] \leq \left(\frac{1}{1 + \frac{\theta}{2}}\right)^{\mathbb{E}[\hat{q}]\theta/2}$$

*Proof.* We will use the following theorem for this proof [26, 27].

**Theorem** (Linial [26])**.** *Let $X_1, \ldots, X_{\hat{p}}$ be Bernoulli $0/1$ random variables. Let $\beta \in (0,1)$ be such that $\beta\hat{p}$ is a positive integer and let $k$ be any positive integer such that $0 < k < \beta\hat{p}$. Then*

$$\mathbb{P}\left[\sum_{i=1}^{\hat{p}} X_i \geq \beta\hat{p}\right] \leq \frac{1}{\binom{\beta\hat{p}}{k}} \sum_{|A|=k} \mathbb{P}\left[\wedge_{i\in A}(X_i = 1)\right]$$

Let $\beta\hat{p} = \mathbb{E}[\hat{q}](1+\theta)$. Now, $\mathbb{P}[X_i = 1] = \mathbb{E}[X_i] = \mathbb{E}[\hat{q}]/\hat{p}$. We will show that

$$\mathbb{P}\left[\wedge_{i\in A}(X_i = 1)\right] \leq (\mathbb{E}[\hat{q}]/\hat{p})^k$$

where $A \subseteq \{1, \ldots, \hat{p}\}$ of size $k$. To see this, note that for any $i$, $\mathbb{P}[X_i = 1] = \mathbb{E}[\hat{q}]/\hat{p}$. The conditional probability of some other $X_j$ being 1 given that $X_i$ is 1 would only reduce. Formally, for $i \neq j$,

$$\mathbb{P}[X_j = 1 | X_i = 1] \leq \mathbb{P}[X_i = 1] = \epsilon\gamma.$$

Note that for $X_i$ to be 1, the Byzantine machines in the $i$-th block must be in the majority. Hence, the reduction in the pool of leftover Byzantine machines was more than honest machines. Since the total number of Byzantine machines is less than the number of honest machines, the probability for them being in a majority in block $j$ reduces. Therefore,

$$
\begin{aligned}
\mathbb{P}\left[\sum_{i=1}^{\hat{p}} X_i \geq \mathbb{E}[\hat{q}](1+\theta)\right] &\leq \frac{\binom{\hat{p}}{k}}{\binom{\mathbb{E}[\hat{q}](1+\theta)}{k}}\mathbb{P}\left[\wedge_{i\in A}(X_i = 1)\right] \\
&\leq \frac{\binom{\hat{p}}{k}}{\binom{\mathbb{E}[\hat{q}](1+\theta)}{k}}(\mathbb{E}[\hat{q}]/\hat{p})^k \\
&\leq \frac{(\hat{p})^k}{k!\binom{\mathbb{E}[\hat{q}](1+\theta)}{k}}\left(\frac{\mathbb{E}[\hat{q}]}{\hat{p}}\right)^k
\end{aligned}
$$

Letting $k = \mathbb{E}[\hat{q}]\theta/2$, we then have

$$
\begin{aligned}
\mathbb{P}\left[\sum_{i=1}^{\hat{p}} X_i \geq \mathbb{E}[\hat{q}](1+\theta)\right] &\leq \frac{(\hat{p})^k}{(\mathbb{E}[\hat{q}](1+\theta/2))^k}(\mathbb{E}[\hat{q}]/\hat{p})^k \\
&= \left(\frac{1}{1+\frac{\theta}{2}}\right)^{\mathbb{E}[\hat{q}]\theta/2}
\end{aligned}
$$

$\square$

## A.4    Proof of Theorem 3

We will adapt the techniques of Theorem 3.1 in [20].

**Lemma 6** ([20], Lemma 2)**.** *Let $\mathbb{H}$ be some Hilbert space, and for $x_1, \ldots, x_k \in \mathbb{H}$, let $x_{gm}$ be their geometric median. Fix $\alpha \in (0, \frac{1}{2})$ and suppose that $z \in \mathbb{H}$ satisfies $\|x_{gm} - z\| > C_\alpha\rho$, where*

$$C_\alpha = (1-\alpha)\sqrt{\frac{1}{1-2\alpha}}$$

*and $\rho > 0$. Then there exists $J \subseteq \{1, \ldots, k\}$ with $|J| > \alpha k$ such that for all $j \in J$, $\|x_j - z\| > \rho$.*

Note that for a general Hilbert or Banach space $\mathbb{H}$, the geometric median is defined as:

$$x_{gm} := \arg\min \sum_{j=1}^{k} \|x - x_j\|_{\mathbb{H}}$$

where $\|.\|_{\mathbb{H}}$ is the norm on $\mathbb{H}$. This coincides with the notion of geometric median in $\mathbb{R}^2$ under the $\ell_2$ norm. Note that Coordinatewise Median is the Geometric Median in the real space with the $\ell^1$ norm, which forms a Banach space.

Firstly, we use Corollary 2 to see that with probability $1 - \delta$, $\hat{q} \leq 1 + 2\log(1/\delta)$. Now, we assume that $\hat{q} \leq 1 + 2\log(1/\delta)$ is true. We will show the remainder of the theorem holds with probability at least $1 - \delta$, as then a union bound will give us the desired result.

**(1):** Let us assume that number of clusters is $k = 128\log 1/\delta$ for some $\delta < 1$, also note that $128\log 1/\delta \geq 8\hat{q}$. Now, choose $\alpha = 1/4$. Choose $\rho = 4\sigma\sqrt{\dfrac{k}{b}}$. Assume that the Geometric Median is more than $C_\alpha \rho$ distance away from true mean. Then by the previous Lemma, atleast $\alpha = 1/4$ fraction of the empirical means of the clusters must lie atleast $\rho$ distance away from true mean. Because we assume the number of clusters is more than $8\hat{q}$, atleast $1/8$ fraction of empirical means of uncorrupted clusters must also lie atleast $\rho$ distance away from true mean.

Recall that the variance of the mean of an "honest" vote group is given by
$$(\sigma')^2 = \sigma^2 \frac{k}{b}.$$

By applying Chebyshev's inequality to the $i^{th}$ uncorrupted vote group $G[i]$, we find that its empirical mean $\hat{x}$ satisfies
$$\mathbb{P}\left(\|G[i] - G\| \geq 4\sigma\sqrt{\frac{k}{b}}\right) \leq \frac{1}{16}.$$

Now, we define a Bernoulli event that is 1 if the empirical mean of an uncorrupted vote group is at distance larger than $\rho$ to the true mean, and 0 otherwise. By the computation above, the probability of this event is less than $1/16$. Thus, its mean is less than $1/16$ and we want to upper bound the probability that empirical mean is more than $1/8$. Using the number of events as $k = 128\log(1/\delta)$, we find that this holds with probability at least $1 - \delta$. For this, we used the following version of Hoeffding's inequality in this part and part (3) of this proof. For Bernoulli events with mean $\mu$, empirical mean $\hat{\mu}$, number of events $m$ and deviation $\theta$:
$$\mathbb{P}(\hat{\mu} - \mu \geq \theta) \leq \exp(-2m\theta^2)$$

To finish the proof, just plug in the values of $C_\alpha$ given in the Lemma 2.1 (written above) from [20], where $C_\alpha = 3/2\sqrt{2}$ for Geometric Median.

**(2):** For coordinate-wise median, we set $k = 128\log d/\delta$. Then we apply the result proved in previous part for each dimension of $\hat{G}$. Then, we get that with probability at least $1 - \delta/d$,
$$|\hat{G}_i - G_i| \leq C_1\sigma_i\sqrt{\frac{\log d/\delta}{b}}$$
where $\hat{G}_i$ is the $i^{th}$ coordinate of $\hat{G}$, $G_i$ is the $i^{th}$ coordinate of $G$ and $\sigma_i^2$ is the $i^{th}$ diagonal entry of $\Sigma$. Doing a union bound, we get that with probability at least $1 - \delta/d$
$$\|\hat{G} - G\| \leq C_1\sigma\sqrt{\frac{\log d/\delta}{b}}.$$

**(3):** Define
$$\Delta_i = \sigma_i\sqrt{\frac{k}{b\sqrt{\frac{1}{2k}\log\frac{d}{\delta}}}}$$
where $\sigma_i^2$ is the $i^{th}$ diagonal entry of $\Sigma$. Now, for each uncorrupted vote group, using Chebyshev's inequality:
$$\mathbb{P}\left(|\hat{G}_i - G_i| \geq \Delta_i\right) \leq \sqrt{\frac{1}{2k}\log\frac{d}{\delta}}.$$

Now, $i^{th}$ coordinate of $\alpha$-trimmed mean lies $\Delta_i$ away from $G_i$ if atleast $\alpha k$ of the $i^{th}$ coordinates of vote group empirical means lie $\Delta_i$ away from $G_i$. Note that because of the assumption of the Proposition $\alpha k \geq 2\hat{q}$. Because $\hat{q}$ of these can be corrupted, atleast $\alpha k/2$ of true empirical means

have $i^{th}$ coordinates that lie $\Delta_i$ away from $G_i$. This means $\alpha/2$ fraction have true empirical means have $i^{th}$ coordinates that lie $\Delta_i$ away from $G_i$. Define a Bernoulli variable $X$ for a vote group as being 1 if the $i^{th}$ coordinate of empirical mean of that vote group lies more than $\Delta_i$ away from $G_i$, and 0 otherwise.

The mean of $X$ therefore satisfies

$$\mathbb{E}(X) < \sqrt{\frac{1}{2k} \log \frac{d}{\delta}}.$$

Set

$$\alpha = 4\sqrt{\frac{1}{2k} \log \frac{d}{\delta}}.$$

Again, using Hoeffding's inequality in a manner analogous to part (1) of the proof, we get that probability of $i^{th}$ coordinate of $\alpha$-trimmed mean being more than $\Delta_i$ away from $G_i$ is less than $\delta/d$.

Taking union bound over all $d$ coordinates, we find that the probability of $\alpha$-trimmed mean being more than

$$\sigma \sqrt{\frac{k}{b\sqrt{\frac{1}{2k} \log \frac{d}{\delta}}}} = \sigma \sqrt{\frac{4k}{b\alpha}}$$

away from $G$ is less than $\delta$. Hence we have proved that if

$$\alpha = 4\sqrt{\frac{1}{2k} \log \frac{d}{\delta}}$$

and $\alpha k \geq 2\hat{q}$, then with probability at least $1 - \delta$, $\Delta \leq \sigma \sqrt{\frac{4k}{b\alpha}}$. Now, set $\alpha = 1/4$ and $k = 128 \log(d/\delta)$. One can easily see that $\alpha k \geq 2\hat{q}$ is satisfied and we get that with probability at least $1 - \delta$, for some constant $C_3$,

$$\Delta \leq C_3 \sigma \sqrt{\frac{\log(d/\delta)}{b}}.$$

## B  Extra Experimental Details

### B.1  Implementation and system-level optimization details

We introduce the details of combining BULYAN, MULTI-KRUM, and coordinate-wise median with DETOX.

- BULYAN: according to [10] BULYAN requires $p \geq 4q + 3$. In DETOX, after the first majority voting level, the corresponding requirement in BULYAN becomes $\frac{p}{r} \geq 4\hat{q} + 3 = 11$. Thus, we assign all "winning" gradients in to one cluster *i.e.*, BULYAN is conducted across 15 gradients.

- MULTI-KRUM: according to [1], MULTI-KRUM requires $p \geq 2q + 3$. Therefore, for similar reason, we assign 15 "winning" gradients into two groups with uneven sizes at 7 and 8 respectively.

- coordinate-wise median: for this baseline we follow the theoretical analysis in Section 3.1 *i.e.*, 15 "winning" gradients are evenly assigned to 5 clusters with size at 3 for *reverse gradient* Byzantine attack. For ALIE attack, we assign those 15 gradients evenly to 3 clusters with size of 5. The reason for this choice is simply that we observe the reported strategies perform better in our experiments. Then mean of the gradients is calculated in each cluster. Finally, we take coordinate-wise median across means of all clusters.

**System-level optimization**  One important thing to point out is that we conducted system level optimizations on implementing MULTI-KRUM and BULYAN, *e.g.*, parallelizing the computationally heavy parts in order to make the comparisons more fair according to [28]. The main idea of our system-level optimization are two-fold: i) gradients of all layers of a neural network are firstly vectorized and concatenated to a high dimensional vector. Robust aggregations are then applied on those high dimensional gradient vectors from all compute nodes. ii) As computational heavy parts

exist for several methods *e.g.*, calculating medians in the second stage of BULYAN. To optimize that part, we chunk the high dimensional gradient vectors evenly into pieces, and parallelize the median calculations in all the pieces. Our system-level optimization leads to 2-4 × speedup in the robust aggregation stage

## B.2  Hyper-parameter tuning

Table 2: Tuned stepsize schedules for experiments under *reverse gradient* Byzantine attack

| Experiments | CIFAR-10 on ResNet-18 | CIFAR-100 on VGG13-BN |
| --- | --- | --- |
| D-MULTI-KRUM | 0.1 | 0.1 |
| D-BULYAN | 0.1 | 0.1 |
| D-Med. | $0.1 \times 0.99^{t \; (\mathrm{mod} \; 10)}$ | $0.1 \times 0.99^{t \; (\mathrm{mod} \; 10)}$ |
| MULTI-KRUM | 0.03125 | 0.03125 |
| BULYAN | 0.1 | 0.1 |
| Med. | 0.1 | $0.1 \times 0.995^{t \; (\mathrm{mod} \; 10)}$ |

Table 3: Tuned stepsize schedules for experiments under ALIE Byzantine attack

| Experiments | CIFAR-10 on ResNet-18 | CIFAR-100 on VGG13-BN |
| --- | --- | --- |
| D-MULTI-KRUM | $0.1 \times 0.98^{t \; (\mathrm{mod} \; 10)}$ | $0.1 \times 0.965^{t \; (\mathrm{mod} \; 10)}$ |
| D-BULYAN | $0.1 \times 0.99^{t \; (\mathrm{mod} \; 10)}$ | $0.1 \times 0.965^{t \; (\mathrm{mod} \; 10)}$ |
| D-Med. | $0.1 \times 0.98^{t \; (\mathrm{mod} \; 10)}$ | $0.1 \times 0.98^{t \; (\mathrm{mod} \; 10)}$ |
| MULTI-KRUM | $0.0078125 \times 0.96^{t \; (\mathrm{mod} \; 10)}$ | $0.00390625 \times 0.965^{t \; (\mathrm{mod} \; 10)}$ |
| BULYAN | $0.001953125 \times 0.95^{t \; (\mathrm{mod} \; 10)}$ | $0.00390625 \times 0.965^{t \; (\mathrm{mod} \; 10)}$ |
| Med. | $0.001953125 \times 0.95^{t \; (\mathrm{mod} \; 10)}$ | $0.001953125 \times 0.965^{t \; (\mathrm{mod} \; 10)}$ |

## B.3  Data augmentation and normalization details

In preprocessing the images in CIFAR-10/100 datasets, we follow the standard data augmentation and normalization process. For data augmentation, random cropping and horizontal random flipping are used. Each color channels are normalized with mean and standard deviation by $\mu_r = 0.491372549, \mu_g = 0.482352941, \mu_b = 0.446666667, \sigma_r = 0.247058824, \sigma_g = 0.243529412, \sigma_b = 0.261568627$. Each channel pixel is normalized by subtracting the mean value in this color channel and then divided by the standard deviation of this color channel.

## B.4  Additional details of the comparison between DETOX and SIGNSGD experiment

**Motivation for the *constant Byzantine attack*** As is argued in [19], the gradient distribution for many modern deep networks can be close to unimodal and symmetric, hence a random sign flip attack is weak since it will not hurt the gradient distribution. And it was shown in the experiments in [19] that SIGNSGD with majority defend the flip sign attack well. We thus consider a stronger while simple *constant Byzantine attack* introduced in Section 5.2 to simulate a more challenging Byzantine distributed training environment. Our expectation is that under this attack, and specifically for SIGNSGD, the Byzantine gradients will mislead model updates towards wrong directions and corrupt the final model trained via SIGNSGD.

**Implementation and hyper-parameter details** We provide more details on our implementation of pairing DETOX with SIGNSGD. To pair DETOX with SIGNSGD, after the majority voting stage of DETOX, we set both $\mathcal{A}_0$ and $\mathcal{A}_1$ as coordinate-wise majority vote describe in Algorithm 1 in [19]. For hyper-parameter tuning, we follow the suggestion in [19] and set the initial learning rate at $0.0001$. However, in defensing the our proposed *constant Byzantine attack*, we observe that constant learning rates lead to model divergence. Thus, we tune the learning rate schedule and use $0.0001 \times 0.99^{t \pmod{10}}$ for both DETOX and DETOX paired SIGNSGD.

## B.5 Mean estimation on synthetic data

Figure 7: Experiment with synthetic data for robust mean estimation: error is reported against dimension (lower is better)

To verify our theoretical analysis, we finally conduct an experiment for a simple mean estimation task. The result of our synthetic mean experiment are shown in Figure 7. In the synthetic mean experiment, we set $p = 220000, r = 11, q = \lfloor \frac{e^r}{3} \rfloor$, and for dimension $d \in \{20, 30, \cdots, 100\}$, we generate 20 samples *iid* from $\mathcal{N}(0, I_d)$. The Byzantine nodes, instead send a constant vector of the same dimension with $\ell_2$ norm of 100. The robustness of an estimator is reflected in the $\ell_2$ norm of its mean estimate. Our experimental results show that DETOX increases the robustness of geometric median and coordinate-wise median, and decreases the dependence of the error on $d$.

## B.6 Effect of varying number of Byzantine nodes

We also study the performance of all considered methods when $q$ (*i.e.* the number of Byzantine nodes) is small and when there is no Byzantine node in the distributed systems. We show here (in Figure 8) the experimental results of $q = 0$ and $q = 1$ (under ALIE Byzantine attack). We observe that DETOX paired versions of robust aggregators consistently beat their standard versions. Different values of $q$ do not seem to affect the robustness and scalability of DETOX.

(a) $q = 1$, VGG13-BN, ALIE attack     (b) $q = 0$, VGG13-BN

Figure 8: Comparison of DETOX paired with BULYAN, MULTI-KRUM versus their vanilla variants for (a) the ALIE attack on VGG13-BN and CIFAR-100 and (b) $q = 0$ (no failures.

## B.7 Comparison between DETOX and DRACO

We provide the experimental results in comparing DETOX with DRACO.

Figure 9: Convergence with respect to runtime comparisons among DETOX back-ended robust aggregation methods and DRACO under *reverse gradient* Byzantine attack on different dataset and model combinations: (a) ResNet-18 trained on CIFAR-10 dataset; (b) VGG13-BN trained on CIFAR-100 dataset

Figure 10: Convergence with respect to runtime comparisons among DETOX back-ended robust aggregation methods and DRACO under *reverse gradient* Byzantine attack on different dataset and model combinations: (a) ResNet-18 trained on CIFAR-10 dataset; (b) VGG13-BN trained on CIFAR-100 dataset.