[Reviews · NeurIPS 2019]

Reviewer 1



My main comment is on the applicability of such robust algorithms in real world scenarios. Can or should the adversarial cases listed in the paper, e.g., power outages and communication failures be modeled as *worst* case attacks? In the seminal work by Lamport the motivation for such worst case attacks are clear. Can the authors show simulations practical cases failures in distributed nodes can be modeled as Byzantine attacks or elaborate more to this point in the paper?

Reviewer 2



This paper presents DETOX, a new framework for improving resistance to Byzantine compute nodes during distributed optimization. The framework utilizes existing robust aggregation techniques, as well as redundancy, within the proposed framework to make better use of resources toward mitigating the effects of Byzantine compute nodes. The paper is very well written. Presentation of existing methods and description of their framework is very clear. The algorithm is presented clearly with minimal assumptions on readers knowledge of distributed training. The theory is sound and shows improvements in performance over multiple relevant metrics. The framework is, from a software design point of view, substantially more complex (cyclomatic complexity may capture this observation) than existing methods and may make adoption of the method more difficult for certain system architectures. The method appears to do quite well when q=5 number of compute nodes are Byzantine, but how does performance suffer when using DETOX if epsilon is small (epsilon=q=0, q=1)? How do other methods (including no robustness and just redundancy) do outside the DETOX framework when epsilon is small? It may make sense to adopt simpler robust methods when epsilon is small. This is minimally addressed with experiments that employ the weaker reverse gradient attack, but having some experiments with varying epsilon may help system developers/architects identify whether DETOX is the right framework for their infrastructure/existing system. Additional thoughts after authors response: I'm a bit confused by their response regarding variance. For p=45, for vanilla aggregation, the PS will average over 45 batches of size b, giving an "effective" batchsize of 45*b, whereas with DETOX, due to redundancy (r=3), we'll see an "effective" batchsize of 15*b. Additional thoughts having gone over the paper again: 1. I think the paper could use more clarity around how their method compares to other methods in terms of resource requirements. It's mentioned in passing (section 5.1 and 5.3) but I think it could be made a bit more explicit. In particular, in section 5.1, do they consider an r=3 group of nodes a compute node? Is this why DETOX computes 3 times as many gradients as the compute nodes in the vanilla aggregation case? 2. Greater clarity around how DETOX doesn't increase variance (for fixed p). My current understanding: For p=45, for vanilla aggregation, the PS will average over 45 batches of size b, giving an "effective" batchsize of 45*b, whereas with DETOX, due to redundancy (r=3), we'll see an "effective" batchsize of 15*b. 3.The paper could benefit from illustrating the problem (Byzantine node failures) with concrete examples (in particular the adversarial setting, unless this is purely hypothetical). I also think this would help make the work more interesting to a wider audience.

Reviewer 3



Originality/significance: The idea of hierarchy in workers (majority vote, then aggregation, then robust aggregation) as detailed in figure 1 is attractive as it only adds a linera (in the number of base nodes) overhead. However, the use of a majority vote in the base layer might lead to a big loss in terms of variance reduction, compared to performing robust aggregation straight from the base layer. Quality/clarity: the paper is fairly well-written and is well suited for the larger audience of NeurIPS

[Author Response · NeurIPS 2019]

1 We thank all reviewers for their overwhelmingly positive feedback on our work. Each reviewer provided helpful
2 suggestions to improve our manuscript that we address below, while providing extra experiments as requested.

**Reviewer 1**
- *"Can or should the adversarial cases listed in the paper [...] be modeled as \*worst\* case attacks?"*

Our work complements a recent, growing body of work on Byzantine ML, where worst-case failures capture a range
of things that can go wrong during training: power outages, software bugs, bit-flips at the storage/network/app level,
and adversarial nodes that corrupt the trained model by sending erroneous gradients. Due to the wide range of failures,
modeling them as worst-case allows for universal robustness guarantees.

- *"Can the authors show simulations practical cases failures [...]?"*

Simulating many different types of failures is interesting but challenging from a system and cost-of-experiments
perspective. Still, in our experiments on real distributed systems, we simulate the strongest known type of node
failures/adversarial gradients, in order to showcase our performance even under the most challenging setups. Under all
these setups, DETOX consistently improves robustness and speed by orders of magnitude.

- *"[...] how their approach is exactly affecting the communication and computation cost [...]?"*

Our communication cost is identical to the vanilla parameter server aggregation cost, as each node sends to the PS
a single gradient. In terms of the cost of computation, we discuss in the paragraph "Improved speed" ln. 160 - 170,
how DETOX improves the aggregation runtime to nearly linear per iteration, cutting down the quadratic runtimes of
state-of-the-art robust aggregators. This improvement naturally varies with different aggregators used, as we discuss in
the same section.

**Reviewer 2**
- *Typos and clarifying variable names.*

Typos fixed. We will restate variable names when it is not clear from context.

- *"The framework is [...] substantially more complex and may make adoption [...] more difficult."*

This is a valid concern. We want to note that DETOX is modular and hardcoded
to the training process. From a user's point-of-view, the only choice required is
what the local aggregators $\mathcal{A}_0$ and $\mathcal{A}_1$ will be. In our implementation (anonymously
available at: `http://bit.ly/2SRyvcS`) this can be done by changing one line of
the code. Since this is a relatively minor code change, we hope that this will make
adoption easier.

- *"Provide [...] results [...] for more values of q, including q=0."*

We will provide a thorough study on the effect of varying $q$ in the camera-ready
version, including the ones shown in Figure 1. Due to the space limit, we show here
the experimental results of $q = 0$ and $q = 1$ (under ALIE Byzantine attack). We
observe that DETOX versions of robust aggregators consistently beat their standard
versions. Different values of $q$ do not seem to affect the robustness and scalability
of DETOX.

(a) $q = 1$, VGG13-BN, ALIE attack

(b) $q = 0$, VGG13-BN

Figure 1: Comparison of DETOX paired with
BULYAN, MULTI-KRUM versus their vanilla variants
for (a) the ALIE attack on VGG13-BN and CIFAR-
100 and (b) $q = 0$ (no failures

**Reviewer 3**
- *"[...] majority vote [...] might lead to a big loss in terms of variance reduction."*

This is a subtle point that can cause confusion. DETOX makes nodes evaluate
redundant gradients, so that there is no increase in variance. Notice that DETOX
first assigns a set of $br/p$ data points to each node group. The nodes in each
group are assigned the same set of $br/p$ points. The nodes then compute the
*mean* of gradients of these points. All "honest" workers in a group return the
same averaged gradient, while averaging leads to variance reduction by a factor
of $br/p$. If the majority is won by the "honest" nodes in the group, this reduced
variance gradient is propagated to the second phase of hierarchical aggregation.
We clarify this in lines 172-176, and this fact is used in the proof of Theorem 3.

- *"[...] I would highly encourage the authors to try incorporating something like signSGD [...] in the base layer."*

Thank you for the suggestion! We agree that incorporating DETOX with
SIGNSGD is valuable. We conducted experiments on DETOX paired with
SIGNSGD versus vanilla SIGNSGD under a *constant* Byzantine attack, where
Byzantine nodes send a constant gradient matrix where all elements equal to
$-1$. The experimental setup is $p = 45, q = 5$. The results are shown in Figure
2. We will include a longer version of this experiment in any camera-ready
version.

(a) ResNet-18 on CIFAR-10

(b) VGG13-BN on CIFAR-100

Figure 2: Convergence of SIGNSGD with and with-
out DETOX under *constant* gradient attack for: (a)
ResNet-18 on CIFAR-10; (b) VGG13-BN on CIFAR-
100

[Meta-Review · NeurIPS 2019]

All the reviewers agreed on the fact that the paper is novel, interesting, and worth to be published in NeurIPS. Please take into account the detailed comments of the reviewers in preparing the camera-ready version.